# Variational Message Passing with Structured Inference Networks

**Wu Lin**[*]**, Nicolas Hubacher**[*]**, Mohammad Emtiyaz Khan** [*]

RIKEN Center for Adavanced Intelligene Project, Tokyo, Japan

wlin2018@cs.ubc.ca, nicolas.hubacher@outlook.com, emtiyaz@gmail.com

## Abstract

Recent efforts on combining deep models with probabilistic graphical models are promising in providing flexible models that are also easy to interpret. We propose a variational message-passing algorithm for variational inference in such models. We make three contributions. First, we propose structured inference networks that incorporate the structure of the graphical model in the inference network of variational auto-encoders (VAE). Second, we establish conditions under which such inference networks enable fast amortized inference similar to VAE. Finally, we derive a variational message passing algorithm to perform efficient natural-gradient inference while retaining the efficiency of the amortized inference. By simultaneously enabling structured, amortized, and natural-gradient inference for deep structured models, our method simplifies and generalizes existing methods.

## 1 Introduction

To analyze real-world data, machine learning relies on models that can extract useful patterns. Deep Neural Networks (DNNs) are a popular choice for this purpose because they can learn flexible representations. Another popular choice are probabilistic graphical models (PGMs) which can find interpretable structures in the data. Recent work on combining these two types of models hopes to exploit their complimentary strengths and provide powerful models that are also easy to interpret (Johnson et al., 2016; Krishnan et al., 2015; Archer et al., 2015; Fraccaro et al., 2016).

To apply such hybrid models to real-world problems, we need efficient algorithms that can extract useful structure from the data. However, the two fields of deep learning and PGMs traditionally use different types of algorithms. For deep learning, stochastic-gradient methods are the most popular choice, e.g., those based on back-propagation. These algorithms are not only widely applicable, but can also employ amortized inference to enable fast inference at test time (Rezende et al., 2014; Kingma & Welling, 2013). On the other hand, most popular algorithms for PGMs exploit the model's graphical conjugacy structure to gain computational efficiency, e.g., variational message passing (VMP) (Winn & Bishop, 2005), expectation propagation (Minka, 2001), Kalman filtering (Ghahramani & Hinton, 1996; 2000), and more recently natural-gradient variational inference (Honkela et al., 2011) and stochastic variational inference (Hoffman et al., 2013). In short, the two fields of deep learning and probabilistic modelling employ fundamentally different inferential strategies and a natural question is, whether we can design algorithms that combine their respective strengths.

There have been several attempts to design such methods in the recent years, e.g., Krishnan et al. (2015; 2017); Fraccaro et al. (2016); Archer et al. (2015); Johnson et al. (2016); Chen et al. (2015). Our work in this paper is inspired by the previous work of Johnson et al. (2016) that aims to combine message-passing, natural-gradient, and amortized inference. Our proposed method in this paper simplifies and generalizes the method of Johnson et al. (2016).

To do so, we propose Structured Inference Networks (SIN) that incorporate the PGM structure in the standard inference networks used in variational auto-encoders (VAE) (Kingma & Welling, 2013; Rezende et al., 2014). We derive conditions under which such inference networks can enable fast amortized inference similar to VAE. By using a recent VMP method of Khan & Lin (2017), we

---

[*]Equal contributions. Wu Lin is now at the University of British Columbia, Vancouver, Canada.

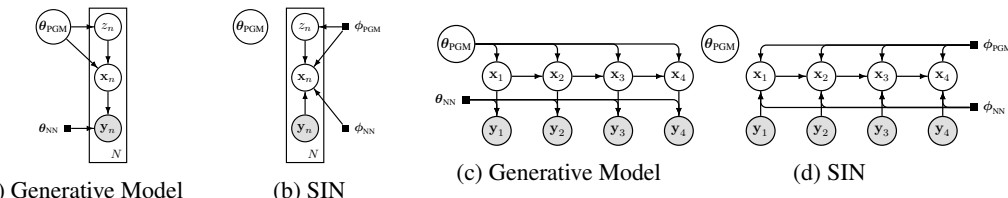

Figure 1: Fig. (a) and (c) show two examples of generative models that combine deep models with PGMs, while Fig. (b) and (d) show our proposed Structured Inference Networks (SIN) for the two models. The generative models are just like the decoder in VAE but they employ a structured prior, e.g., Fig. (a) has a mixture-model prior while Fig. (b) has a dynamical system prior. SINs, just like the encoder in VAE, mimic the structure of the generative model by using parameters $\phi$. One main difference is that in SIN the arrows between $\mathbf{y}_n$ and $\mathbf{x}_n$ are reversed compared to the model, while rest of the arrows have the same direction.

derive a variational message-passing algorithm whose messages automatically reduce to stochastic-gradients for the deep components of the model, while perform natural-gradient updates for the PGM part. Overall, our algorithm enables Structured, Amortized, and Natural-gradient (SAN) updates and therefore we call our algorithm the SAN algorithm. We show that our algorithm give comparable performance to the method of Johnson et al. (2016) while simplifying and generalizing it. The code to reproduce our results is available at **https://github.com/emtiyaz/vmp-for-svae/**.

## 2 THE MODEL AND CHALLENGES WITH ITS INFERENCE

We consider the modelling of data vectors $\mathbf{y}_n$ by using local latent vectors $\mathbf{x}_n$. Following previous works (Johnson et al., 2016; Archer et al., 2015; Krishnan et al., 2015), we model the output $\mathbf{y}_n$ given $\mathbf{x}_n$ using a neural network with parameters $\boldsymbol{\theta}_{\mathrm{NN}}$, and capture the correlations among data vectors $\mathbf{y} := \{\mathbf{y}_1, \mathbf{y}_2, \ldots, \mathbf{y}_N\}$ using a probabilistic graphical model (PGM) over the latent vectors $\mathbf{x} := \{\mathbf{x}_1, \mathbf{x}_2, \ldots, \mathbf{x}_N\}$. Specifically, we use the following joint distribution:

$$p(\mathbf{y}, \mathbf{x}, \boldsymbol{\theta}) := \underbrace{\left[ \prod_{n=1}^{N} p(\mathbf{y}_n | \mathbf{x}_n, \boldsymbol{\theta}_{\mathrm{NN}}) \right]}_{\text{DNN}} \underbrace{\left[ p(\mathbf{x} | \boldsymbol{\theta}_{\mathrm{PGM}}) \right]}_{\text{PGM}} \underbrace{\left[ p(\boldsymbol{\theta}_{\mathrm{PGM}}) \right]}_{\text{Hyperprior}}, \tag{1}$$

where $\boldsymbol{\theta}_{\mathrm{NN}}$ and $\boldsymbol{\theta}_{\mathrm{PGM}}$ are parameters of a DNN and PGM respectively, and $\boldsymbol{\theta} := \{\boldsymbol{\theta}_{\mathrm{NN}}, \boldsymbol{\theta}_{\mathrm{PGM}}\}$.

This combination of probabilistic graphical model and neural network is referred to as structured variational auto-encoder (SVAE) by Johnson et al. (2016). SVAE employs a structured prior $p(\mathbf{x} | \boldsymbol{\theta}_{\mathrm{PGM}})$ to extract useful structure from the data. SVAE therefore differs from VAE (Kingma & Welling, 2013) where the prior distribution over $\mathbf{x}$ is simply a multivariate Gaussian distribution $p(\mathbf{x}) = \mathcal{N}(\mathbf{x} | 0, \mathbf{I})$ with no special structure. To illustrate this difference, we now give an example.

**Example (Mixture-Model Prior):** Suppose we wish to group the outputs $\mathbf{y}_n$ into $K$ distinct clusters. For such a task, the standard Gaussian prior used in VAE is not a useful prior. We could instead use a mixture-model prior over $\mathbf{x}_n$, as suggested by (Johnson et al., 2016),

$$p(\mathbf{x} | \boldsymbol{\theta}_{\mathrm{PGM}}) = \prod_{n=1}^{N} p(\mathbf{x}_n | \boldsymbol{\theta}_{\mathrm{PGM}}) = \prod_{n=1}^{N} \left[ \sum_{k=1}^{K} p(\mathbf{x}_n | z_n = k) \pi_k \right], \tag{2}$$

where $z_n \in \{1, 2, \ldots, K\}$ is the mixture indicator for the $n$'th data example, and $\pi_k$ are mixing proportions that sum to 1 over $k$. Each mixture component can further be modelled, e.g., by using a Gaussian distribution $p(\mathbf{x}_n | z_n = k) := \mathcal{N}(\mathbf{x}_n | \boldsymbol{\mu}_k, \boldsymbol{\Sigma}_k)$ giving us the Gaussian Mixture Model (GMM) prior with PGM hyperparameters $\boldsymbol{\theta}_{\mathrm{PGM}} := \{\boldsymbol{\mu}_k, \boldsymbol{\Sigma}_k, \pi_k\}_{k=1}^{K}$. The graphical model of an SVAE with such priors is shown in Figure 1a. This type of structured-prior is useful for discovering clusters in the data, making them easier to interpret than VAE.

Our main goal in this paper is to approximate the posterior distribution $p(\mathbf{x}, \boldsymbol{\theta}|\mathbf{y})$. Specifically, similar to VAE, we would like to approximate the posterior of $\mathbf{x}$ by using an inference network. In VAE, this is done by using a function parameterized by DNN, as shown below:

$$p(\mathbf{x}|\mathbf{y}, \boldsymbol{\theta}_{\mathrm{NN}}) = \frac{1}{p(\mathbf{y}|\boldsymbol{\theta})} \prod_{n=1}^{N} [p(\mathbf{y}_n|\mathbf{x}_n, \boldsymbol{\theta}_{\mathrm{NN}})\mathcal{N}(\mathbf{x}_n|0, \mathbf{I})] \approx \prod_{n=1}^{N} q(\mathbf{x}_n|f_\phi(\mathbf{y}_n)), \qquad (3)$$

where the left hand side is the posterior distribution of $\mathbf{x}$, and the first equality is obtained by using the distribution of the decoder in the Bayes' rule. The right hand side is the distribution of the encoder where $q$ is typically an exponential-family distribution whose natural-parameters are modelled by using a DNN $f_\phi$ with parameters $\phi$. The same function $f_\phi(\cdot)$ is used for all $n$ which reduces the number of variational parameters and enables sharing of statistical strengths across $n$. This leads to both faster training and faster testing (Rezende et al., 2014).

Unfortunately, for SVAE, such inference networks may give inaccurate predictions since they ignore the structure of the PGM prior $p(\mathbf{x}|\boldsymbol{\theta}_{\mathrm{PGM}})$. For example, suppose $\mathbf{y}_n$ is a time-series and we model $\mathbf{x}_n$ using a dynamical system as depicted in Fig. 1c. In this case, the inference network of (3) is not an accurate approximation since it ignores the time-series structure in $\mathbf{x}$. This might result in inaccurate predictions of distant future observations, e.g., prediction for an observation $\mathbf{y}_{10}$ given the past data $\{\mathbf{y}_1, \mathbf{y}_2, \mathbf{y}_3\}$ would be inaccurate because the inference network has no path connecting $\mathbf{x}_{10}$ to $\mathbf{x}_1, \mathbf{x}_2$, or $\mathbf{x}_3$. In general, whenever the prior structure is important in obtaining accurate predictions, we might want to incorporate it in the inference network.

A solution to this problem is to use an inference network with the same structure as the model but to replace all its edges by neural networks (Krishnan et al., 2015; Fraccaro et al., 2016). This solution is reasonable when the PGM itself is complex, but might be too aggressive when the PGM is a simple model, e.g., when the prior in Fig. 1c is a linear dynamical system. Using DNNs in such cases would dramatically increase the number of parameters which will lead to a possible deterioration in both speed and performance.

Johnson et al. (2016) propose a method to incorporate the structure of the PGM part in the inference network. For SVAE with conditionally-conjugate PGM priors, they aim to obtain a mean-field variational inference by optimizing the following standard variational lower bound[1]:

$$\mathcal{L}(\boldsymbol{\lambda}_x, \boldsymbol{\theta}) := \mathbb{E}_{q(x)} \left[ \log \left\{ \prod_{n=1}^{N} p(\mathbf{y}_n|\mathbf{x}_n, \boldsymbol{\theta}_{\mathrm{NN}})p(\mathbf{x}|\boldsymbol{\theta}_{\mathrm{PGM}}) \right\} - \log q(\mathbf{x}|\boldsymbol{\lambda}_x) \right], \qquad (4)$$

where $q(\mathbf{x}|\boldsymbol{\lambda}_x)$ is a minimal exponential-family distribution with natural parameters $\boldsymbol{\lambda}_x$. To incorporate an inference network, they need to restrict the parameter of $q(\mathbf{x}|\boldsymbol{\lambda}_x)$ similar to the VAE encoder shown in (3), i.e., $\boldsymbol{\lambda}_x$ must be defined using a DNN with parameter $\phi$. For this purpose, they use a two-stage iterative procedure. In the first stage, they obtain $\boldsymbol{\lambda}_x^*$ by optimizing a surrogate lower bound where the decoder in (4) is replaced by the VAE encoder of (3) (highlighted in blue),

$$\hat{\mathcal{L}}(\boldsymbol{\lambda}_x, \boldsymbol{\theta}, \phi) := \mathbb{E}_{q(x)} \left[ \log \left\{ \prod_{n=1}^{N} q(\mathbf{x}_n|f_\phi(\mathbf{y}_n))p(\mathbf{x}|\boldsymbol{\theta}_{\mathrm{PGM}}) \right\} - \log q(\mathbf{x}|\boldsymbol{\lambda}_x) \right]. \qquad (5)$$

The optimal $\boldsymbol{\lambda}_x^*$ is a function of $\boldsymbol{\theta}$ and $\phi$ and they denote it by $\boldsymbol{\lambda}_x^*(\boldsymbol{\theta}, \phi)$. In the second stage, they substitute $\boldsymbol{\lambda}_x^*$ into (4) and take a gradient step to optimize $\mathcal{L}(\boldsymbol{\lambda}_x^*(\boldsymbol{\theta}, \phi), \boldsymbol{\theta})$ with respect to $\boldsymbol{\theta}$ and $\phi$. This is iterated until convergence. The first stage ensures that $q(\mathbf{x}|\boldsymbol{\lambda}_x)$ is defined in terms of $\phi$ similar to VAE, while the second stage improves the lower bound while maintaining this restriction.

The advantage of this formulation is that when the factors $q(\mathbf{x}_n|f_\phi(\mathbf{y}_n))$ are chosen to be conjugate to $p(\mathbf{x}|\boldsymbol{\theta}_{\mathrm{PGM}})$, the first stage can be performed efficiently using VMP. However, the overall method might be difficult to implement and tune. This is because the procedure is equivalent to an implicitly-constrained optimization[2] that optimizes (4) with the constraint $\boldsymbol{\lambda}_x^*(\boldsymbol{\theta}, \phi) = \arg\max_{\boldsymbol{\lambda}_x} \hat{\mathcal{L}}(\boldsymbol{\lambda}_x, \boldsymbol{\theta}, \phi)$. Such constrained problems are typically more difficult to solve than their unconstrained counterparts, especially when the constraints are nonconvex (Heinkenschloss, 2008). Theoretically, the convergence of such methods is difficult to guarantee when the constraints are

---

[1]Johnson et al. (2016) consider $\boldsymbol{\theta}$ to be a random variable, but for clarity we assume $\boldsymbol{\theta}$ to be deterministic.
[2]This is similar to Hoffman & Blei (2015) who also solve an implicitly constrained optimization problem.

violated. In practice, this makes the implementation difficult because in every iteration the VMP updates need to run long enough to reach close to a local optimum of the surrogate lower bound.

Another disadvantage of the method of Johnson et al. (2016) is that its efficiency could be ensured only under restrictive assumptions on the PGM prior. For example, the method does not work for PGMs that contain non-conjugate factors because in that case VMP cannot be used to optimize the surrogate lower bound. In addition, the method is not directly applicable when $\boldsymbol{\lambda}_x$ is constrained and when $p(\mathbf{x}|\boldsymbol{\theta}_{\text{PGM}})$ has additional latent variables (e.g., indicator variables $z_n$ in the mixture-model example). In summary, the method of Johnson et al. (2016) might be difficult to implement and tune, and also difficult to generalize to cases when PGM is complex.

In this paper, we propose an algorithm to simplify and generalize the algorithm of Johnson et al. (2016). We propose structured inference networks (SIN) that incorporate the structure of the PGM part in the VAE inference network. Even when the graphical model contains a non-conjugate factor, SIN can preserve some structure of the model. We derive conditions under which SIN can enable efficient amortized inference by using stochastic gradients. We discuss many examples to illustrate the design of SIN for many types of PGM structures. Finally, we derive a VMP algorithm to perform natural-gradient variational inference on the PGM part while retaining the efficiency of the amortized inference on the DNN part.

## 3 STRUCTURED INFERENCE NETWORKS

We start with the design of inference networks that incorporate the PGM structure into the inference network of VAE. We propose the following *structured* inference network (SIN) which consists of two types of factors,

$$q(\mathbf{x}|\mathbf{y}, \boldsymbol{\phi}) := \frac{1}{\mathcal{Z}(\boldsymbol{\phi})} \underbrace{\left[\prod_{n=1}^{N} q(\mathbf{x}_n | f_{\phi_{\text{NN}}}(\mathbf{y}_n))\right]}_{\text{DNN Factor}} \underbrace{\left[q(\mathbf{x}|\boldsymbol{\phi}_{\text{PGM}})\right]}_{\text{PGM Factor}}. \tag{6}$$

The DNN factor here is similar to (3) while the PGM factor is an exponential-family distribution which has a similar graph structure as the PGM prior $p(\mathbf{x}|\boldsymbol{\theta}_{\text{PGM}})$. The role of the DNN term is to enable flexibility while the role of the PGM term is to incorporate the model's PGM structure into the inference network. Both factors have their own parameters. $\boldsymbol{\phi}_{\text{NN}}$ is the parameter of DNN and $\boldsymbol{\phi}_{\text{PGM}}$ is the natural parameter of the PGM factor. The parameter set is denoted by $\boldsymbol{\phi} := \{\boldsymbol{\phi}_{\text{NN}}, \boldsymbol{\phi}_{\text{PGM}}\}$.

How should we choose the two factors? As we will show soon that, for fast amortized inference, these factors need to satisfy the following two conditions. The first condition is that the normalizing constant[3] $\log \mathcal{Z}(\boldsymbol{\phi})$ is easy to evaluate and differentiate. The second condition is that we can draw samples from SIN, i.e., $\mathbf{x}^*(\boldsymbol{\phi}) \sim q(\mathbf{x}|\mathbf{y}, \boldsymbol{\phi})$ where we have denoted the sample by $\mathbf{x}^*(\boldsymbol{\phi})$ to show its dependence on $\boldsymbol{\phi}$. An additional desirable, although not necessary, feature is to be able to compute the gradient of $\mathbf{x}^*(\boldsymbol{\phi})$ by using the reparameterization trick. Now, we will show that given these two conditions we can easily perform amortized inference.

We show that when the above two conditions are met, a stochastic gradient of the lower bound can be computed in a similar way as in VAE. For now, we assume that $\boldsymbol{\theta}$ is a deterministic variable (we will relax this in the next section). The variational lower bound in this case can be written as follows:

$$\mathcal{L}_{\text{SIN}}(\boldsymbol{\theta}, \boldsymbol{\phi}) := \mathbb{E}_q \left[\log \frac{p(\mathbf{y}, \mathbf{x}|\boldsymbol{\theta})}{q(\mathbf{x}|\mathbf{y}, \boldsymbol{\phi})}\right] = \mathbb{E}_q \left[\log \frac{\prod_n \{p(\mathbf{y}_n|\mathbf{x}_n, \boldsymbol{\theta}_{\text{NN}})\} p(\mathbf{x}|\boldsymbol{\theta}_{\text{PGM}})\mathcal{Z}(\boldsymbol{\phi})}{\prod_n \{q(\mathbf{x}_n|f_{\phi_{\text{NN}}}(\mathbf{y}_n))\} q(\mathbf{x}|\boldsymbol{\phi}_{\text{PGM}})}\right] \tag{7}$$

$$= \sum_{n=1}^{N} \mathbb{E}_q \left[\log \frac{p(\mathbf{y}_n|\mathbf{x}_n, \boldsymbol{\theta}_{\text{NN}})}{q(\mathbf{x}_n|f_{\phi_{\text{NN}}}(\mathbf{y}_n))}\right] + \mathbb{E}_q[\log p(\mathbf{x}|\boldsymbol{\theta}_{\text{PGM}})] - \mathbb{E}_q[\log q(\mathbf{x}|\boldsymbol{\phi}_{\text{PGM}})] + \log \mathcal{Z}(\boldsymbol{\phi}) \tag{8}$$

The first term above is identical to the lower bound of the standard VAE, while the rest of the terms are different (shown in blue). The second term differs due to the PGM prior in the generative model. In VAE, $p(\mathbf{x}|\boldsymbol{\theta}_{\text{PGM}})$ is a standard normal, but here it is a structured PGM prior. The last two terms arise due to the PGM term in SIN. If we can compute the gradients of the last three terms and generate samples $\mathbf{x}^*(\boldsymbol{\phi})$ from SIN, we can perform amortized inference similar to VAE. Fortunately,

---

[3]Note that both the factors are normalized distribution, but their product may not be.

the second and third terms are usually easy for PGMs, therefore we only require the gradient of $\mathcal{Z}(\phi)$ to be easy to compute. This confirms the two conditions required for a fast amortized inference.

The resulting expressions for the stochastic gradients are shown below where we highlight in blue the additional gradient computations required on top of a VAE implementation (we also drop the explicit dependence of $\mathbf{x}^*(\phi)$ over $\phi$ for notational simplicity).

$$\frac{\partial \mathcal{L}_{\text{SIN}}}{\partial \theta_{\text{NN}}} \approx N \frac{\partial \log p(\mathbf{y}_n | \mathbf{x}_n^*, \boldsymbol{\theta}_{\text{NN}})}{\partial \theta_{\text{NN}}}, \qquad \frac{\partial \mathcal{L}_{\text{SIN}}}{\partial \theta_{\text{PGM}}} \approx \frac{\partial \log p(\mathbf{x}^* | \boldsymbol{\theta}_{\text{PGM}})}{\partial \theta_{\text{PGM}}}, \qquad (9)$$

$$\frac{\partial \mathcal{L}_{\text{SIN}}}{\partial \phi} \approx \frac{\partial}{\partial \mathbf{x}_n^*} \left[ N \log \frac{p(\mathbf{y}_n | \mathbf{x}_n^*, \boldsymbol{\theta}_{\text{NN}})}{q(\mathbf{x}_n^* | f_{\phi_{\text{NN}}}(\mathbf{y}_n))} \right] \frac{\partial \mathbf{x}_n^*}{\partial \phi} - N \frac{\partial \log q(\mathbf{x}_n^* | f_{\phi_{\text{NN}}}(\mathbf{y}_n))}{\partial \phi}$$

$$+ \frac{\partial}{\partial \mathbf{x}^*} \left[ \log \frac{p(\mathbf{x}^* | \boldsymbol{\theta}_{\text{PGM}})}{q(\mathbf{x}^* | \phi_{\text{PGM}})} \right] \frac{\partial \mathbf{x}^*}{\partial \phi} + \frac{\partial \log \mathcal{Z}(\phi)}{\partial \phi}, \qquad (10)$$

The gradients of $\mathcal{Z}(\phi)$ and $\mathbf{x}^*(\phi)$ might be cheap or costly depending on the type of PGM. For example, for LDS, these require a full inference through the model which costs $O(N)$ computation and is infeasible for large $N$. However, for GMM, each $\mathbf{x}_n$ can be independently sampled and therefore computations are independent of $N$. In general, if the latent variables in PGM are highly correlated (e.g., Gaussian process prior), then Bayesian inference is not computationally efficient and gradients are difficult to compute. In this paper, we do not consider such difficult cases and assume that $\mathcal{Z}(\phi)$ and $\mathbf{x}^*(\phi)$ can be evaluated and differentiated cheaply.

We now give many examples of SIN that meet the two conditions required for a fast amortized inference. When $p(\mathbf{x} | \boldsymbol{\theta}_{\text{PGM}})$ is a conjugate exponential-family distribution, choosing the two factors is a very easy task. In this case, we can let $q(\mathbf{x} | \phi_{\text{PGM}}) = p(\mathbf{x} | \phi_{\text{PGM}})$, i.e., the second factor is the same distribution as the PGM prior but with a different set of parameters $\phi_{\text{PGM}}$. To illustrate this, we give an example below when the PGM prior is a linear dynamical system.

**Example (SIN for Linear Dynamical System (LDS)) :** When $\mathbf{y}_n$ is a time series, we can model the latent $\mathbf{x}_n$ using an LDS defined as $p(\mathbf{x} | \boldsymbol{\theta}) := \mathcal{N}(\mathbf{x}_0 | \boldsymbol{\mu}_0, \boldsymbol{\Sigma}_0) \prod_{n=1}^{N} \mathcal{N}(\mathbf{x}_n | \mathbf{A}\mathbf{x}_{n-1}, \mathbf{Q})$, where $\mathbf{A}$ is the transition matrix, $\mathbf{Q}$ is the process-noise covariance, and $\boldsymbol{\mu}_0$ and $\boldsymbol{\Sigma}_0$ are the mean and covariance of the initial distribution. Therefore, $\boldsymbol{\theta}_{\text{PGM}} := \{\mathbf{A}, \mathbf{Q}, \boldsymbol{\mu}_0, \boldsymbol{\Sigma}_0\}$. In our inference network, we choose $q(\mathbf{x} | \phi_{\text{PGM}}) = p(\mathbf{x} | \phi_{\text{PGM}})$ as show below, where $\phi_{\text{PGM}} := \{\bar{\mathbf{A}}, \bar{\mathbf{Q}}, \bar{\boldsymbol{\mu}}_0, \bar{\boldsymbol{\Sigma}}_0\}$ and, since our PGM is a Gaussian, we choose the DNN factor to be a Gaussian as well:

$$q(\mathbf{x} | \mathbf{y}, \phi) := \frac{1}{\mathcal{Z}(\phi)} \underbrace{\left[ \prod_{n=1}^{N} \mathcal{N}(\mathbf{x}_n | \mathbf{m}_n, \mathbf{V}_n) \right]}_{\text{DNN Factor}} \underbrace{\left[ \mathcal{N}(\mathbf{x}_0 | \bar{\boldsymbol{\mu}}_0, \bar{\boldsymbol{\Sigma}}_0) \prod_{n=1}^{N} \mathcal{N}(\mathbf{x}_n | \bar{\mathbf{A}}\mathbf{x}_{n-1}, \bar{\mathbf{Q}}) \right]}_{\text{LDS Factor}}, \quad (11)$$

where $\mathbf{m}_n := \mathbf{m}_{\phi_{\text{NN}}}(\mathbf{y}_n)$ and $\mathbf{V}_n := \mathbf{V}_{\phi_{\text{NN}}}(\mathbf{y}_n)$ are mean and covariance parameterized by a DNN with parameter $\phi_{\text{NN}}$. The generative model and SIN are shown in Fig. 1c and 1d, respectively. The above SIN is a conjugate model where the marginal likelihood and distributions can be computed in $O(N)$ using the forward-backward algorithm, a.k.a. Kalman smoother (Bishop, 2016). We can also compute the gradient of $\mathcal{Z}(\phi)$ as shown in Kokkala et al. (2015).

When the PGM prior has additional latent variables, e.g., the GMM prior has cluster indicators $z_n$, we might want to incorporate their structure in SIN. This is illustrate in the example below.

**Example (SIN for GMM prior):** The prior shown in (2) has an additional set of latent variables $z_n$. To mimic this structure in SIN, we choose the PGM factor as shown below with parameters $\phi_{\text{PGM}} := \{\bar{\boldsymbol{\mu}}_k, \bar{\boldsymbol{\Sigma}}_k, \bar{\pi}_k\}_{k=1}^{K}$, while keeping the DNN part to be a Gaussian distribution similar to the LDS case:

$$q(\mathbf{x} | \mathbf{y}, \phi) := \frac{1}{\mathcal{Z}(\phi)} \prod_{n=1}^{N} \underbrace{\left[ \mathcal{N}(\mathbf{x}_n | \mathbf{m}_n, \mathbf{V}_n) \right]}_{\text{DNN Factor}} \underbrace{\left[ \sum_{k=1}^{K} \mathcal{N}(\mathbf{x}_n | \bar{\boldsymbol{\mu}}_k, \bar{\boldsymbol{\Sigma}}_k) \bar{\pi}_k \right]}_{\text{GMM Factor}}, \qquad (12)$$

The model and SIN are shown in Figure 1a and 1b, respectively. Fortunately, due to conjugacy of the Gaussian and multinomial distributions, we can marginalize $\mathbf{x}_n$ to get a closed-form expression for $\log \mathcal{Z}(\boldsymbol{\phi}) := \sum_n \log \sum_k \mathcal{N}(\mathbf{m}_n | \bar{\boldsymbol{\mu}}_k, \mathbf{V}_n + \bar{\boldsymbol{\Sigma}}_k) \, \bar{\pi}_k$. We can sample from SIN by first sampling from the marginal $q(z_n = k | \mathbf{y}, \boldsymbol{\phi}) \propto \mathcal{N}(\mathbf{m}_n | \bar{\boldsymbol{\mu}}_k, \mathbf{V}_n + \bar{\boldsymbol{\Sigma}}_k) \, \bar{\pi}_k$. Given $z_n$, we can sample $\mathbf{x}_n$ from the following conditional: $q(\mathbf{x}_n | z_n = k, \mathbf{y}, \boldsymbol{\phi}) = \mathcal{N}(\mathbf{x}_n | \widetilde{\boldsymbol{\mu}}_n, \widetilde{\boldsymbol{\Sigma}}_n)$, where $\widetilde{\boldsymbol{\Sigma}}_n^{-1} = \mathbf{V}_n^{-1} + \bar{\boldsymbol{\Sigma}}_k^{-1}$ and $\widetilde{\boldsymbol{\mu}}_n = \widetilde{\boldsymbol{\Sigma}}_n (\mathbf{V}_n^{-1} \mathbf{m}_n + \bar{\boldsymbol{\Sigma}}_k^{-1} \boldsymbol{\mu}_k)$. See Appendix B for a detailed derivation.

In all of the above examples, we are able to satisfy the two conditions even when we use the same structure as the model. However, this may not always be possible for all conditionally-conjugate exponential family distributions. However, we can still obtain samples from a tractable structured mean-field approximation using VMP. We illustrate this for the switching state-space model in Appendix A. In such cases, a drawback of our method is that we need to run VMP long enough to get a sample, very similar to the method of Johnson et al. (2016). However, our gradients are simpler to compute than theirs. Their method requires gradients of $\boldsymbol{\lambda}^*(\boldsymbol{\theta}, \boldsymbol{\phi})$ which depends both on $\boldsymbol{\theta}$ and $\boldsymbol{\phi}$ (see Proposition 4.2 in (Johnson et al., 2016)). In our case, we require gradient of $\mathcal{Z}(\boldsymbol{\phi})$ which is independent of $\boldsymbol{\theta}$ and therefore is simpler to implement.

An advantage of our method over the method of Johnson et al. (2016) is that our method can handle non-conjugate factors in the generative model. When the PGM prior contains some non-conjugate factors, we might replace them by their closest conjugate approximations while making sure that the inference network captures the useful structure present in the posterior distribution. We illustrate this on a Student's t mixture model.

**Example (SIN for Student's t-Mixture Model) :** To handle outliers in the data, we might want to use the Student's t-mixture component in the mixture model shown in (2), i.e., we set $p(\mathbf{x}_n | z_n = k) = \mathcal{T}(\mathbf{x}_n | \boldsymbol{\mu}_k, \boldsymbol{\Sigma}_k, \boldsymbol{\gamma}_k)$ with mean $\boldsymbol{\mu}_k$, scale matrix $\boldsymbol{\Sigma}_k$ and degree of freedom $\boldsymbol{\gamma}_k$. The Student's t-distribution is not conjugate to the multinomial distribution, therefore, if we use it as the PGM factor in SIN, we will not be able to satisfy both conditions easily. Even though our model contains a t-distribution components, we can still use the SIN shown in (12) that uses a GMM factor. We can therefore simplify inference by choosing an inference network which has a simpler form than the original model.

In theory, one can do this even when all factors are non-conjugate, however, the approximation error might be quite large in some cases for this approximation to be useful. In our experiments, we tried this for non-linear dynamical system and found that capturing non-linearity was essential for dynamical systems that are extremely non-linear.

## 4  VARIATIONAL MESSAGE PASSING FOR NATURAL-GRADIENT VARIATIONAL INFERENCE

Previously, we assumed $\boldsymbol{\theta}_{\text{PGM}}$ to be deterministic. In this section, we relax this condition and assume $\boldsymbol{\theta}_{\text{PGM}}$ to follow an exponential-family prior $p(\boldsymbol{\theta}_{\text{PGM}} | \boldsymbol{\eta}_{\text{PGM}})$ with natural parameter $\boldsymbol{\eta}_{\text{PGM}}$. We derive a VMP algorithm to perform natural-gradient variational inference for $\boldsymbol{\theta}_{\text{PGM}}$. Our algorithm works even when the PGM part contains non-conjugate factors, and it does not affect the efficiency of the amortized inference on the DNN part. We assume the following mean-field approximation: $q(\mathbf{x}, \boldsymbol{\theta} | \mathbf{y}) := q(\mathbf{x} | \mathbf{y}, \boldsymbol{\phi}) q(\boldsymbol{\theta}_{\text{PGM}} | \boldsymbol{\lambda}_{\text{PGM}})$ where the first term is equal to SIN introduced in the previous section, and the second term is an exponential-family distribution with natural parameter $\boldsymbol{\lambda}_{\text{PGM}}$. For $\boldsymbol{\theta}_{\text{NN}}$ and $\boldsymbol{\phi}$, we will compute point estimates.

We build upon the method of Khan & Lin (2017) which is a generalization of VMP and stochastic variational inference (SVI). This method enables natural-gradient updates even when PGM contains non-conjugate factors. This method performs natural-gradient variational inference by using a mirror-descent update with the Kullback-Leibler (KL) divergence. To obtain natural-gradients with respect to the natural parameters of $q$, the mirror-descent needs to be performed in the mean parameter space. We will now derive a VMP algorithm using this method.

We start by deriving the variational lower bound. The variational lower bound corresponding to the mean-field approximation can be expressed in terms of $\mathcal{L}_{\text{SIN}}$ derived in the previous section.

$$\mathcal{L}(\boldsymbol{\lambda}_{\text{PGM}}, \boldsymbol{\theta}_{\text{NN}}, \boldsymbol{\phi}) := \mathbb{E}_{q(\theta_{\text{PGM}} | \lambda_{\text{PGM}})} [\mathcal{L}_{\text{SIN}}(\boldsymbol{\theta}, \boldsymbol{\phi})] - \mathbb{D}_{KL}[q(\boldsymbol{\theta}_{\text{PGM}} | \boldsymbol{\lambda}_{\text{PGM}}) \| p(\boldsymbol{\theta}_{\text{PGM}} | \boldsymbol{\eta}_{\text{PGM}})]. \quad (13)$$

---

**Algorithm 1** Structured, Amortized, and Natural-gradient (SAN) Variational Inference

---

**Require:** Data $\mathbf{y}$, Step-sizes $\beta_1, \beta_2, \beta_3$
1: Initialize $\boldsymbol{\lambda}_{\text{PGM}}, \boldsymbol{\theta}_{\text{NN}}, \boldsymbol{\phi}$.
2: **repeat**
3:      Compute $q(\mathbf{x}|\mathbf{y}, \boldsymbol{\phi})$ for SIN shown in (6) either by using an exact expression or using VMP.
4:      Sample $\mathbf{x}^* \sim q(\mathbf{x}|\mathbf{y}, \boldsymbol{\phi})$, and compute $\nabla_{\boldsymbol{\phi}} \mathcal{Z}$ and $\nabla_{\boldsymbol{\phi}} \mathbf{x}^*$.
5:      Update $\boldsymbol{\lambda}_{\text{PGM}}$ using the natural-gradient step given in (16).
6:      Update $\boldsymbol{\theta}_{\text{NN}}$ and $\boldsymbol{\phi}$ using the gradients given in (9)-(10) with $\boldsymbol{\theta}_{\text{PGM}} \sim q(\boldsymbol{\theta}_{\text{PGM}}|\boldsymbol{\lambda}_{\text{PGM}})$.
7: **until** Convergence

---

We will use a mirror-descent update with the KL divergence for $q(\boldsymbol{\theta}_{\text{PGM}}|\boldsymbol{\lambda}_{\text{PGM}})$ because we want natural-gradient updates for it. For the rest of the parameters, we will use the usual Euclidean distance. We denote the mean parameter corresponding to $\boldsymbol{\lambda}_{\text{PGM}}$ by $\boldsymbol{\mu}_{\text{PGM}}$. Since $q$ is a minimal exponential family, there is a one-to-one map between the mean and natural parameters, therefore we can reparameterize $q$ such that $q(\boldsymbol{\theta}_{\text{PGM}}|\boldsymbol{\lambda}_{\text{PGM}}) = q(\boldsymbol{\theta}_{\text{PGM}}|\boldsymbol{\mu}_{\text{PGM}})$. Denoting the values at iteration $t$ with a superscript $t$ and using Eq. 19 in (Khan & Lin, 2017) with these divergences, we get:

$$\max_{\mu_{\text{PGM}}} \ \langle \boldsymbol{\mu}_{\text{PGM}}, \nabla_{\mu_{\text{PGM}}} \mathcal{L}_t \rangle - \frac{1}{\beta_1} \mathbb{D}_{KL}[q(\boldsymbol{\theta}_{\text{PGM}}|\boldsymbol{\mu}_{\text{PGM}}) \| q(\boldsymbol{\theta}_{\text{PGM}}|\boldsymbol{\mu}_{\text{PGM}}^t)], \tag{14}$$

$$\max_{\theta_{\text{NN}}} \ \langle \boldsymbol{\theta}_{\text{NN}}, \nabla_{\theta_{\text{NN}}} \mathcal{L}_t \rangle - \frac{1}{\beta_2} \|\boldsymbol{\theta}_{\text{NN}} - \boldsymbol{\theta}_{\text{NN}}^t\|_2^2, \qquad \max_{\phi} \ \langle \boldsymbol{\phi}, \nabla_{\phi} \mathcal{L}_t \rangle - \frac{1}{\beta_3} \|\boldsymbol{\phi} - \boldsymbol{\phi}^t\|_2^2. \tag{15}$$

where $\beta_1$ to $\beta_3$ are scalars, $\langle, \rangle$ is an inner product, and $\nabla \mathcal{L}_t$ is the gradient at the value in iteration $t$.

As shown by Khan & Lin (2017), the maximization in (14) can be obtained in closed-form:

$$\boldsymbol{\lambda}_{\text{PGM}} \leftarrow (1 - \beta_1) \boldsymbol{\lambda}_{\text{PGM}} + \beta_1 \nabla_{\mu_{\text{PGM}}} \mathbb{E}_{q(\theta_{\text{PGM}}|\mu_{\text{PGM}})}[\log p(\mathbf{x}^*|\boldsymbol{\theta}_{\text{PGM}})]. \tag{16}$$

When the prior $p(\boldsymbol{\theta}_{\text{PGM}}|\boldsymbol{\eta}_{\text{PGM}})$ is conjugate to $p(\mathbf{x}|\boldsymbol{\theta}_{\text{PGM}})$, the above step is equal to the SVI update of the global variables. The gradient itself is equal to the message received by $\boldsymbol{\theta}_{\text{PGM}}$ in a VMP algorithm, which is also the natural gradient with respect to $\boldsymbol{\lambda}_{\text{PGM}}$. When the prior is not conjugate, the gradient can be approximated either by using stochastic gradients or by using the reparameterization trick (Khan & Lin, 2017). Therefore, this update enables natural-gradient update for PGMs that may contain both conjugate and non-conjugate factors.

The update of the rest of the parameters can be done by using a stochastic-gradient method. This is because the solution of the update (15) is equal to a stochastic-gradient descent update (one can verify this by simplify taking the gradient and setting it to zero). We can compute the stochastic-gradients by using a Monte Carlo estimate with a sample $\boldsymbol{\theta}_{\text{PGM}}^* \sim q(\boldsymbol{\theta}_{\text{PGM}}|\boldsymbol{\lambda}_{\text{PGM}})$ as shown below:

$$\nabla_{\phi} \mathcal{L}(\boldsymbol{\lambda}_{\text{PGM}}, \boldsymbol{\theta}_{\text{NN}}, \boldsymbol{\phi}) \approx \nabla_{\phi} \mathcal{L}_{\text{SIN}}(\boldsymbol{\theta}^*, \boldsymbol{\phi}), \qquad \nabla_{\theta_{\text{NN}}} \mathcal{L}(\boldsymbol{\lambda}_{\text{PGM}}, \boldsymbol{\theta}_{\text{NN}}, \boldsymbol{\phi}) \approx \nabla_{\theta_{\text{NN}}} \mathcal{L}_{\text{SIN}}(\boldsymbol{\theta}^*, \boldsymbol{\phi}) \tag{17}$$

where $\boldsymbol{\theta}^* := \{\boldsymbol{\theta}_{\text{PGM}}^*, \boldsymbol{\theta}_{\text{NN}}\}$. As discussed in the previous section, these gradients can be computed similar to VAE-like by using the gradients given in (9)-(10). Therefore, for the DNN part we can perform amortized inference, and use a natural-gradient update for the PGM part using VMP.

The final algorithm is outlined in Algorithm 1. Since our algorithm enables Structured, Amortized, and Natural-gradient (SAN) updates, we call it the SAN algorithm. Our updates conveniently separate the PGM and DNN computations. Step 3-6 operate on the PGM part, for which we can use existing implementation for the PGM. Step 7 operates on the DNN part, for which we can reuse VAE implementation. Our algorithm not only generalizes previous works, but also simplifies the implementation by enabling the reuse of the existing software.

## 5    EXPERIMENTS AND RESULTS

The main goal of our experiments is to show that our SAN algorithm gives similar results to the method of Johnson et al. (2016). For this reason, we apply our algorithm to the two examples considered in Johnson et al. (2016), namely the latent GMM and latent LDS (see Fig. 1). In this section we discuss results for latent GMM. An additional result for LDS is included in Appendix C. Our results show that, similar to the method of Johnson et al. (2016) our algorithm can learn complex

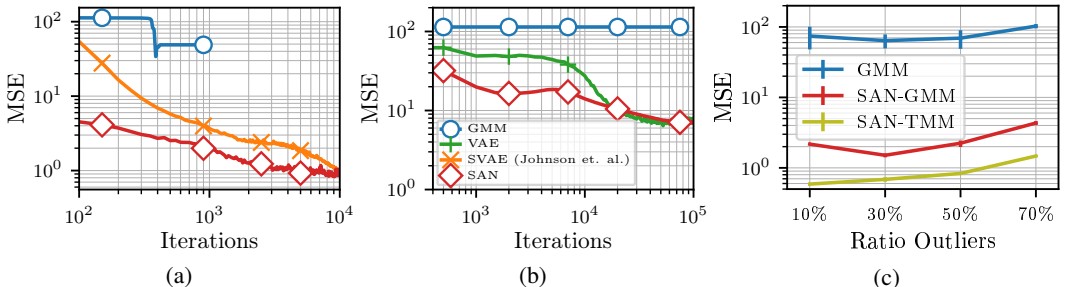

Figure 2: Figure (a) compares performances of GMM, SVAE, and SAN on the Pinwheel where we see that SAN converges faster than SVAE and performs better than GMM. Figure (b) compares GMM, VAE, and SAN on the Auto dataset where we see the same trend. Figure (c) compares performances on the Pinwheel dataset with outliers. We see that the performance of SAN on Student's t-mixture model (SAN-TMM) degrades slower than the performance of methods based on GMM. Even with 70% outliers, SAN-TMM performs better than SAN-GMM with 10% outliers.

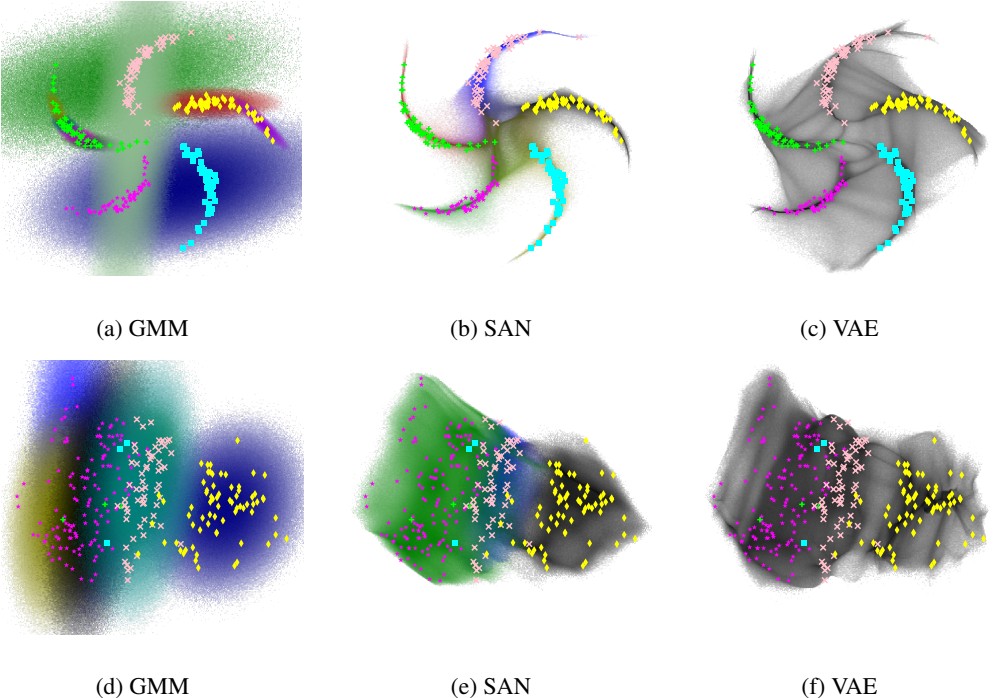

Figure 3: Top row is for the Pinwheel dataset, while the bottom row is for the Auto dataset. Point clouds in the background of each plot show the samples generated from the learned generative model, where each mixture component is shown with a different color and the color intensities are proportional to the probability of the mixture component. The points in the foreground show data samples which are colored according to the true labels. We use $K = 10$ mixture components to train all models. For the Auto dataset, we show only the first two principle components.

representations with interpretable structures. The advantage of our method is that it is simpler and more general than the method of Johnson et al. (2016).

We compare to three baseline methods. The first method is the variational expectation-maximization (EM) algorithm applied to the standard Gaussian mixture model. We refer to this method as 'GMM'. This method is a clustering method but does not use a DNN to do so. The second method is the VAE approach of Kingma & Welling (2013), which we refer to as 'VAE'. This method uses a DNN but does not cluster the outputs or latent variables. The third method is the SVAE approach of Johnson

et al. (2016) applied to latent GMM shown in Fig. 1. This method uses both a DNN and a mixture model to cluster the latent variables. We refer to this as 'SVAE'. We compare these methods to our SAN algorithm applied to latent GMM model. We refer to our method as 'SAN'. All methods employ a Normal-Wishart prior over the GMM hyperparameters (see Bishop (2016) for details).

We use two datasets. The first dataset is the synthetic two-dimensional Pinwheel dataset ($N = 5000$ and $D = 2$) used in (Johnson et al., 2016). The second dataset is the Auto dataset ($N = 392$ and $D = 6$, available in the UCI repository) which contains information about cars. The dataset also contains a five-class label which indicates the number of cylinders in a car. We use these labels to validate our results. For both datasets we use $70\%$ data for training and the rest for testing. For all methods, we tune the step-sizes, the number of mixture components, and the latent dimensionality on a validation set. We train the GMM baseline using a batch method, and, for VAE and SVAE, we use minibatches of size 64. DNNs in all models consist of two layers with 50 hidden units and an output layer of dimensionality 6 and 2 for the Auto and Pinwheel datasets, respectively.

Figure 2a and 2b compare the performances during training. In Figure 2a, we compare to SVAE and GMM, where we see that SAN converges faster than SVAE. As expected, both SVAE and SAN achieve similar performance upon convergence and perform better than GMM. In Figure 2b, we compare to VAE and GMM, and observe similar trends. The performance of GMM is represented as a constant because it converges after a few iterations already. We found that the implementation provided by Johnson et al. (2016) does not perform well on the Auto dataset which is why we have not included it in the comparison. We also compared the test log-likelihoods and imputation error which show very similar trends. We omit these results due to space constraints.

In the background of each plot in Figure 3, we show samples generated from the generative model. In the foreground, we show the data with the true labels. These labels were not used during training. The plots (a)-(c) show results for the Pinwheel dataset, while plots (d)-(e) shows results for the Auto dataset. For the Auto dataset, each label corresponds to the number of cylinders present in a car. We observe that SAN can learn meaningful clusters of the outputs. On the other hand, VAE does not have any mechanisms to cluster and, even though the generated samples match the data distribution, the results are difficult to interpret. Finally, as expected, both SAN and VAE learn flexible patterns while GMM fails to do so. Therefore, SAN enables flexible models that are also easy to interpret.

An advantage of our method over the method of Johnson et al. (2016) is that our method applies even when PGM contains non-conjugate factors. Now, we discuss a result for such a case. We consider the SIN for latent Student's t-mixture model (TMM) discussed in Section 3. The generative model contains the student's t-distribution as a non-conjugate factor, but our SIN replaces it with a Gaussian factor. When the data contains outliers, we expect the SIN for latent TMM to perform better than the SIN for latent GMM. To show this, we add artificial outliers to the Pinwheel dataset using a Gaussian distribution with a large variance. We fix the degree of freedom for the Student's t-distribution to 5. We test on four different levels of noise and report the test MSE averaged over three runs for each level. Figure 2c shows a comparison of GMM, SAN on latent GMM, and SAN on latent TMM where we see that, as the noise level is increased, latent TMM's performance degrades slower than the other methods (note that the y-axis is in log-scale). Even with $70\%$ of outliers, the latent TMM still performs better than the latent GMM with only $10\%$ of outliers. This experiment illustrates that a conjugate SIN can be used for inference on a model with a non-conjugate factor.

## 6 DISCUSSION AND CONCLUSION

We propose an algorithm to simplify and generalize the algorithm of Johnson et al. (2016) for models that contain both deep networks and graphical models. Our proposed VMP algorithm enables structured, amortized, and natural-gradient updates given that the structured inference networks satisfy two conditions. The two conditions derived in this paper generally hold for PGMs that do not force dense correlations in the latent variables $\mathbf{x}$. However, it is not clear how to extend our method to models where this is the case, e.g., Gaussian process models. It is possible to use ideas from sparse Gaussian process models and we will investigate this in the future. An additional issue is that our results are limited to small scale data. We found that it is non-trivial to implement a message-passing framework that goes well with the deep learning framework. We are going to pursue this direction in the future and investigate good platforms to integrate the capabilities of these two different flavors of algorithms.

**Acknowledgement:** We would like to thank Didrik Nielsen (RIKEN) for his help during this work. We would also like to thank Matthew J. Johnson (Google Brain) and David Duvenaud (University of Toronto) for providing the SVAE code. Finally, we are thankful for the RAIDEN computing system at RIKEN AIP center, which we used for our experiments.

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

## A  SIN FOR SWITCHING LDS

In SLDS, we introduce discrete variable $z_n \in \{1, 2, \ldots, K\}$ that are sampled using a Markov chain: $p(z_n = i | z_{n-1} = j) = \pi_{ij}$ such that $\pi_{ij}$ sum to 1 over all i given j. The transition for LDS is defined conditioned on $z_n$: $p(\mathbf{x}_n | \mathbf{x}_{n-1}, z_n = i, \boldsymbol{\theta}_{\text{PGM}}) := \mathcal{N}(\mathbf{x}_n | \mathbf{A}_i \mathbf{x}_{n-1}, \mathbf{Q}_i)$ where $\mathbf{A}_i$ and $\mathbf{Q}_i$ are parameters for the $i$'th indicator. These two dynamics put together define the SLDS prior $p(\mathbf{x}, \mathbf{z} | \boldsymbol{\theta}_{\text{PGM}})$. We can use the following SIN which uses the SLDS prior as the PGM factor but with parameters $\boldsymbol{\phi}_{\text{PGM}}$ instead of $\boldsymbol{\theta}_{\text{PGM}}$. The expression for $q(\mathbf{x}, \mathbf{z} | \mathbf{y}, \boldsymbol{\phi})$ is shown below:

$$\frac{1}{\mathcal{Z}(\boldsymbol{\phi})} \underbrace{\left[ \prod_{n=1}^{N} \mathcal{N}(\mathbf{x}_n | \mathbf{m}_n, \mathbf{V}_n) \right]}_{\text{DNN Factor}} \underbrace{\left[ \mathcal{N}(\mathbf{x}_0 | \bar{\boldsymbol{\mu}}_{0,z_0}, \bar{\boldsymbol{\Sigma}}_{0,z_0}) \prod_{n=1}^{N} \mathcal{N}(\mathbf{x}_n | \bar{\mathbf{A}}_{z_n} \mathbf{x}_{n-1}, \bar{\mathbf{Q}}_{z_n}) p(z_n | z_{n-1}) \right]}_{\text{SLDS Factor}},$$

Even though the above model is a conditionally-conjugate model, the partition function is not tractable and sampling is also not possible. However, we can use a structured mean-field approximation. First, we can combine the DNN factor with the Gaussian observation of SLDS factor and then use a mean-field approximation $q(\mathbf{x}, \mathbf{z} | \mathbf{y}, \boldsymbol{\phi}) \approx q(\mathbf{x} | \boldsymbol{\lambda}_x) q(\mathbf{z} | \boldsymbol{\lambda}_z)$, e.g., using the method of Ghahramani & Hinton (2000). This will give us a structured approximation where the edges between $\mathbf{y}_n$ and $\mathbf{x}_n$ and $z_n$ and $z_{n-1}$ are maintained but $\mathbf{x}_n$ and $z_n$ independent of each other.

## B  SIN FOR SVAE WITH MIXTURE MODEL PRIOR

In this section we give detailed derivations for the SIN shown in (12). We derive the normalizing constant $\mathcal{Z}(\boldsymbol{\phi})$ and show how to generate samples from SIN.

We start by a simple rearrangement of SIN defined in (12):

$$q(\mathbf{x} | \mathbf{y}, \boldsymbol{\phi}) \propto \prod_{n=1}^{N} \mathcal{N}(\mathbf{x}_n | \mathbf{m}_n, \mathbf{V}_n) \left[ \sum_{k=1}^{K} \mathcal{N}(\mathbf{x}_n | \bar{\boldsymbol{\mu}}_k, \bar{\boldsymbol{\Sigma}}_k) \bar{\pi}_k \right]$$

$$= \prod_{n=1}^{N} \sum_{k=1}^{K} \mathcal{N}(\mathbf{x}_n | \mathbf{m}_n, \mathbf{V}_n) \mathcal{N}(\mathbf{x}_n | \bar{\boldsymbol{\mu}}_k, \bar{\boldsymbol{\Sigma}}_k) \bar{\pi}_k, \tag{18}$$

$$\propto \prod_{n=1}^{N} \sum_{k=1}^{K} q(\mathbf{x}_n, z_n = k | \mathbf{y}_n, \boldsymbol{\phi}) \tag{19}$$

where the first step follows from the definition (12), the second step follows by taking the sum over $k$ outside, and the third step is obtained by defining each component as a joint distribution over $\mathbf{x}_n$ and the indicator variable $z_n$.

We will express this joint distribution as a multiplication of the marginal of $z_n$ and conditional of $\mathbf{x}_n$ given $z_n$. We will see that this will give us the expression for the normalizing constant, as well as a way to sample from SIN.

We can simplify the joint distribution further as shown below. The first step follows from the definition. The second step is obtained by swapping $\mathbf{m}_n$ and $\mathbf{x}_n$ in the first term. The third step is obtained by completing the squares and expressing the first term as a distribution over $\mathbf{x}_n$ (the second and third terms are independent of $\mathbf{x}_n$).

$$q(\mathbf{x}_n, z_n = k | \mathbf{y}_n, \boldsymbol{\phi}) \propto \mathcal{N}(\mathbf{x}_n | \mathbf{m}_n, \mathbf{V}_n) \mathcal{N}(\mathbf{x}_n | \bar{\boldsymbol{\mu}}_k, \bar{\boldsymbol{\Sigma}}_k) \bar{\pi}_k \tag{20}$$

$$= \mathcal{N}(\mathbf{m}_n | \mathbf{x}_n, \mathbf{V}_n) \mathcal{N}(\mathbf{x}_n | \bar{\boldsymbol{\mu}}_k, \bar{\boldsymbol{\Sigma}}_k) \bar{\pi}_k \tag{21}$$

$$= \mathcal{N}(\mathbf{x}_n | \widetilde{\boldsymbol{\mu}}_n, \widetilde{\boldsymbol{\Sigma}}_n) \mathcal{N}(\mathbf{m}_n | \bar{\boldsymbol{\mu}}_k, \mathbf{V}_n + \bar{\boldsymbol{\Sigma}}_k) \bar{\pi}_k, \tag{22}$$

where $\widetilde{\boldsymbol{\Sigma}}_n^{-1} := \mathbf{V}_n^{-1} + \bar{\boldsymbol{\Sigma}}_k^{-1}$ and $\widetilde{\boldsymbol{\mu}}_n := \widetilde{\boldsymbol{\Sigma}}_n \left( \mathbf{V}_n^{-1} \mathbf{m}_n + \bar{\boldsymbol{\Sigma}}_k^{-1} \boldsymbol{\mu}_k \right)$.

Using the above we get the marginal of $z_n$ and conditional of $\mathbf{x}_n$ given $z_n$:

$$q(z_n = k | \mathbf{y}_n, \boldsymbol{\phi}) \propto \mathcal{N}(\mathbf{m}_n | \bar{\boldsymbol{\mu}}_k, \mathbf{V}_n + \bar{\boldsymbol{\Sigma}}_k) \bar{\pi}_k \tag{23}$$

$$q(\mathbf{x}_n | z_n = k, \mathbf{y}_n, \boldsymbol{\phi}) := \mathcal{N}(\mathbf{x}_n | \widetilde{\boldsymbol{\mu}}_n, \widetilde{\boldsymbol{\Sigma}}_n) \tag{24}$$

The normalizing constant of the marginal of $z_n$ is obtained by simply summing over all $k$:

$$\mathcal{Z}_n(\boldsymbol{\phi}) := \sum_{k=1}^{K} \mathcal{N}\left(\mathbf{m}_n | \bar{\boldsymbol{\mu}}_k, \mathbf{V}_n + \bar{\boldsymbol{\Sigma}}_k\right) \bar{\pi}_k. \tag{25}$$

and since $q(\mathbf{x}_n | z_n = k, \mathbf{y}_n, \boldsymbol{\phi})$ is already a normalized distribution, we can write the final expression for the SIN as follows:

$$q(\mathbf{x}|\mathbf{y}, \boldsymbol{\phi}) = \prod_{n=1}^{N} \frac{1}{\mathcal{Z}_n(\boldsymbol{\phi})} \sum_{k=1}^{K} q(\mathbf{x}_n | z_n = k, \mathbf{y}_n, \boldsymbol{\phi}) q(z_n = k | \mathbf{y}_n, \boldsymbol{\phi}) \tag{26}$$

where components are defined in (23),(24), and (25). The normalizing constant is available in closed-form and we can sample $z_n$ first and then generate $\mathbf{x}_n$. This completes the derivation.

## C  RESULTS FOR LATENT LINEAR DYNAMICAL SYSTEM

In this experiment, we apply our SAN algorithm to the latent LDS discussed in Section 3. For comparison, we compare our method, Structured Variational Auto-Encoder (SVAE) (Johnson et al., 2016), and LDS on the Dot dataset used in Johnson et al. (2016). Our results show that our method achieves comparable performance to SVAE. For LDS, we perform batch learning for all model parameters using the EM algorithm. For SVAE and SAN, we perform mini-batch updates for all model parameters. We use the same neutral network architecture as in Johnson et al. (2016), which contains two hidden layers with tanh activation function. We repeat our experiments 10 times and measure model performance in terms of the following mean absolute error for $\tau$-steps ahead prediction. The error measures the absolute difference between the ground truth and the generative outputs by averaging across generated results.

$$\sum_{n=1}^{N} \sum_{t=1}^{T-\tau} \frac{1}{N(T-\tau)d} \left\{ ||\mathbf{y}_{t+\tau,n}^* - \mathbb{E}_{p(y_{t+\tau,n}|y_{1:t,n})} \left[\mathbf{y}_{t+\tau,n}\right] ||_1 \right\} \tag{27}$$

where $N$ is the number of testing time series with $T$ time steps, $d$ is the dimensionality of observation $\mathbf{y}$, and observation $\mathbf{y}_{t+\tau,n}^*$ denotes the ground-truth at time step $t + \tau$.

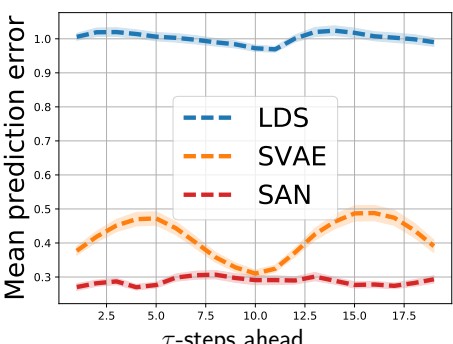

Figure 4: Prediction error, where shadows denote the standard errors across 10 runs

From Figure 4, we can observe that our method performs as good as SVAE and outperforms LDS. Our method is slightly robust than SVAE. In Figure 5, there are generated images obtained from all methods. From Figure 5, we also see that our method performs as good as SAVE and is able to recover the ground-truth observation.

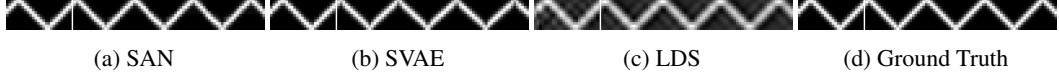

     (a) SAN            (b) SVAE            (c) LDS          (d) Ground Truth

Figure 5: Generated images, where each column of pixels represents an observation, each row of pixels represents one time step, and each vertical white line denotes the first time step to generate images

