# OpenReview forum: "Variational Message Passing with Structured Inference Networks"
_ICLR.cc/2018/Conference — Accept (Poster)_

### Official Review · AnonReviewer1 · 2017-11-24
**More efficient variational inference with structured inference networks**

**Rating:** 7
**Confidence:** 3

**Review:**

The authors adapts stochastic natural gradient methods for variational inference with structured inference networks. The variational approximation proposed is similar to SVAE by Jonhson et al. (2016), but rather than directly using the global variable theta in the local approximation for x the authors propose to optimize a separate variational parameter. The authors then extends and adapts the natural gradient method by Khan & Lin (2017) to optimize all the variational parameters. In the experiments the authors generally show improved convergence over SVAE.

The idea seems promising but it is still a bit unclear to me why removing dependence between global and local parameters that you know is there would lead to a better variational approximation. The main motivation seems to be that it is easier to optimize.

- In the last two sentences of the updates for \theta_PGM you mention that you need to do SVI/VMP to compute the function \eta_x\theta. Might this also suffer from non-convergence issues like you argue SVAE does? Or do you simply mean that computation of this is exact using regular message passing/Kalman filter/forward-backward?
- It was not clear to me why we should use a Gaussian approximation for the \theta_NN parameters? The prior might be Gaussian but the posterior is not? Is this more of a simplifying assumption?
- There has recently been interest in using inference networks as part of more flexible variational approximations for structured models. Some examples of related work missing in this area is "Variational Sequential Monte Carlo" by Naesseth et al. (2017) / "Filtering Variational Objectives" by Maddison et al. (2017) / "Auto-encoding sequential Monte Carlo" Le et al. (2017).
-  Section 2.1, paragraph nr 5, "algorihtm" -> "algorithm"

---

> ### Author Response · Authors · 2017-12-21
> **We will improve the clarity of writing, our motivation, and the description of our method. We will also add 2 new experiments.**
>
> Thanks for your review. Following reviewers suggestions, we will make the following major changes in our next draft:
> - We will clearly explain that our main motivation is to improve over the method of Johnson et.al., 2016.
> - We will clarify the description, especially the conjugacy requirements, and add detailed discussion on the applicability and limitations of our approach.
> - We will clean-up the description of our algorithm in Section 4.
> - We will add an experiment on non-conjugate mixture model to demonstrate the generality of our approach. We will add a larger experiment for clustering MNIST digits.
> --------------
> DETAILED COMMENTS
> Reviewer: “Might this also suffer from non-convergence issues like you argue SVAE does? Or do you simply mean that computation of this is exact using regular message passing/Kalman filter/forward-backward?”
>
> Response: Our method does not have convergence issues, and is guaranteed to converge under mild conditions discussed in Khan and Lin 2017. And yes, the updates are exact and obtained using regular message passing.
> --------------
> Reviewer: “It was not clear to me why we should use a Gaussian approximation for the \theta_NN parameters? “
>
> Response: You are right. We can use any appropriate exponential family approximation. The updates are similar to Khan and Lin’s method. We will change this in the final draft.
> --------------
> Thanks for the citations. We will add them in the paper

---

> ### Author Response · Authors · 2018-01-05
> **Changes in the Revised Version**
>
> We have made the following changes in the revised version:
>  - Introduction is modified to show that our method is an improvement over Johnson et. al.’s method, and it builds upon Khan and Lin’s method.
>  - Section 2 modified to clearly show the issues with Johnson et.al.’s method.
>  - Section 3 modified to clarify the conjugacy requirements of inference network. We have added many illustrative examples.
>  - Section 4 modified to simplify the algorithm description. We have added a pseudo-code.
>  - In Section 5 we added a new result on Student’s-t mixture model.

---

### Official Review · AnonReviewer3 · 2017-11-27
**This paper presents an interesting algorithm to perform variational inference with amortized and natural gradient updates of models that contain deep neural network and probabilistic graphical model components. The contributions of the paper need to be more clearly presented and the experiments could be more thorough.**

**Rating:** 7
**Confidence:** 4

**Review:**

This paper presents a variational inference algorithm for models that contain
deep neural network components and probabilistic graphical model (PGM)
components.
The algorithm implements natural-gradient message-passing where the messages
automatically reduce to stochastic gradients for the non-conjugate neural
network components. The authors demonstrate the algorithm on a Gaussian mixture
model and linear dynamical system where they show that the proposed algorithm
outperforms previous algorithms. Overall, I think that the paper proposes some
interesting ideas, however, in its current form I do not think that the novelty
of the contributions are clearly presented and that they are not thoroughly
evaluated in the experiments.

The authors propose a new variational inference algorithm that handles models
with deep neural networks and PGM components. However, it appears that the
authors rely heavily on the work of (Khan & Lin, 2017) that actually provides
the algorithm. As far as I can tell this paper fits inference networks into
the algorithm proposed in (Khan & Lin, 2017) which boils down to i) using an
inference network to generate potentials for a conditionally-conjugate
distribution and ii) introducing new PGM parameters to decouple the inference
network from the model parameters. These ideas are a clever solution to work
inference networks into the message-passing algorithm of (Khan & Lin, 2017),
but I think the authors may be overselling these ideas as a brand new algorithm.
I think if the authors sold the paper as an alternative to (Johnson, et al., 2016)
that doesn't suffer from the implicit gradient problem the paper would fit into
the existing literature better.

Another concern that I have is that there are a lot of conditiona-conjugacy
assumptions baked into the algorithm that the authors only mention at the end
of the presentation of their algorithm. Additionally, the authors briefly state
that they can handle non-conjugate distributions in the model by just using
conjugate distributions in the variational approximation. Though one could do
this, the authors do not adequately show that one should, or that one can do this
without suffering a lot of error in the posterior approximation. I think that
without an experiment the small section on non-conjugacy should be removed.

Finally, I found the experimental evaluation to not thoroughly demonstrate the
advantages and disadvantages of the proposed algorithm. The algorithm was applied
to the two models originally considered in (Johnson, et al., 2016) and the
proposed algorithm was shown to attain lower mean-square errors for the two
models. The experiments do not however demonstrate why the algorithm is
performing better. For instance, is the (Johnson, et al., 2016) algorithm
suffering from the implicit gradient? It also would have been great to have
considered a model that the (Johnson, et. al., 2016) algorithm would not work
well on or could not be applied to show the added applicability of the proposed
algorithm.

I also have some minor comments on the paper:
- There are a lot of typos.
- The first two sentences of the abstract do not really contribute anything
  to the paper. What is a powerful model? What is a powerful algorithm?
- DNN was used in Section 2 without being defined.
- Using p() as an approximate distribution in Section 3 is confusing notation
  because p() was used for the distributions in the model.
- How is the covariance matrix parameterized that the inference network produces?
- The phrases "first term of the inference network" are not clear. Just use The
  DNN term and the PGM term of the inference networks, and better still throw
  in a reference to Eq. (4).
- The term "deterministic parameters" was used and never introduced.
- At the bottom of page 5 the extension to the non-conjugate case should be
  presented somewhere (probably the appendix) since the fact that you can do
  this is a part of your algorithm that's important.

---

> ### Author Response · Authors · 2017-12-21
> **We will improve the clarity of writing, our motivation, and the description of our method. We will also add 2 new experiments.**
>
> Thanks for the review. Following reviewers suggestions, we will make the following major changes in our next draft:
> - We will clearly explain that our main motivation is to improve over the method of Johnson et.al., 2016.
> - We will clarify the description, especially the conjugacy requirements, and add detailed discussion on the applicability and limitations of our approach.
> - We will clean-up the description of our algorithm in Section 4.
> - We will add an experiment on non-conjugate mixture model to demonstrate the generality of our approach. We will add a larger experiment for clustering MNIST digits.
> -----------
> DETAILED COMMENTS
> Reviewer: “However, it appears that the authors rely heavily on the work of (Khan & Lin, 2017) that actually provides the algorithm. [...] I think the authors may be overselling these ideas as a brand new algorithm.”.
>
> Response: Thanks for letting us know. This was not our intention. We will modify the write-up to clarify our contributions over Khan and Lin, 2017 and not oversell our method.
> -----------
> Reviewer: “I think if the authors sold the paper as an alternative to (Johnson, et al., 2016) that doesn't suffer from the implicit gradient problem the paper would fit into the existing literature better.”
>
> Response: That’s a good point and we will write about this a bit more clearly in the paper. Our method is not just an alternative over Jonson et. al. 2016, but it is a generalization of their method. We propose a Variational Message Passing framework for complex models which are not covered by the method of Johnson et. al. 2016. We will modify the introduction and discussion to clarify these points. We will also add an experiment on a non-conjugate model as an evidence of the generality of our approach.
> -----------
> Reviewer: “Another concern that I have is that there are a lot of conditional-conjugacy assumptions baked into the algorithm that the authors only mention at the end of the presentation of their algorithm.”
>
> Response: We agree and we will clarify the text to reflect the following point: Our method works for general non-conjugate models, but our inference network is restricted to a conjugate model where the normalizing constant is easy to compute.
> -----------
> Reviewer: “The authors briefly state that they can handle non-conjugate distributions in the model [...]. Though one could do this, the authors do not adequately show that one should, or that one can do this without suffering a lot of error in the posterior approximation.”
>
> Response: We will modify the text to clarify this. We will also add an example of a non-conjugate mixture model and show how to design a conjugate inference network for this problem. We will also add a paragraph explaining how to generalize this procedure to general graphical models.
> -----------
> Reviewer: “the experimental evaluation do not thoroughly demonstrate the advantages and disadvantages of the proposed algorithm…. The experiments do not however demonstrate why the algorithm is performing better.”
>
> Response: Thanks for pointing this out. The goal of our experiments was to show that, when PGM is a conjugate model, our method performs similar to the method of Johnson et. al. The advantage of our approach is the simplicity of our method, as well as its generality. This is mentioned in the first paragraph in Section 5.
> -----------
> Reviewer: “is the (Johnson, et al., 2016) algorithm suffering from the implicit gradient?”
>
> Response: On small datasets, we did not observe the implicit gradient issue with the method of Johnson et. al. But in principle we expect this to be a problem for complex models.
> -----------
> Reviewer: “It also would have been great to have considered a model that the (Johnson, et. al., 2016) algorithm would not work well on or could not be applied to show the added applicability of the proposed algorithm.”
>
> Response: Thanks for the suggestion. We will add an experiment for non-conjugate mixture model, where the method of Johnson et. al. does not apply.
> -----------
> Thanks for further suggestions. We will modify the abstract to remove the line about “powerful models and algorithms”. We will take other comments into account as well. Thanks!

---

> ### Author Response · Authors · 2018-01-05
> **Changes in the Revised Version**
>
> We have made the following changes in the revised version:
>  - Introduction is modified to show that our method is an improvement over Johnson et. al.’s method, and it builds upon Khan and Lin’s method.
>  - Section 2 modified to clearly show the issues with Johnson et.al.’s method.
>  - Section 3 modified to clarify the conjugacy requirements of inference network. We have added many illustrative examples.
>  - Section 4 modified to simplify the algorithm description. We have added a pseudo-code.
>  - In Section 5 we added a new result on Student’s-t mixture model.

---

### Official Review · AnonReviewer2 · 2017-11-28
**The paper proposes an approach to perform inference in models that combine probabilistic graphical models with deep networks**

**Rating:** 7
**Confidence:** 2

**Review:**

The paper seems to be significant since it integrates PGM inference with deep models. Specifically, the idea is to use the structure of the PGM to perform efficient inference. A variational message passing approach is developed which performs natural-gradient updates for the PGM part and stochastic gradient updates for the deep model part. Performance comparison is performed with an existing approach that does not utilize the PGM structure for inference.
The paper does a good job of explaining the challenges of inference, and provides a systematic approach to integrating PGMs with deep model updates. As compared to the existing approach where the PGM parameters must converge before updating the DNN parameters, the proposed architecture does not require this, due to the re-parameterization which is an important contribution.

The motivation of the paper, and the description of its contribution as compared to existing methods can be improved. One of the main aspects it seems is generality, but the encodings are specific to 2 types PGMs. Can this be generalized to arbitrary PGM structures? How about cases when computing Z is intractable? Could the proposed approach be adapted to such cases. I was not very sure as to why the proposed method is more general than existing approaches.

Regarding the experiments, as mentioned in the paper the evaluation is performed on two fairly small scale datasets. the approach shows that the proposed methods converge faster than existing methods. However, I think there is value in the approach, and the connection between variational methods with DNNs is interesting.

---

> ### Author Response · Authors · 2017-12-21
> **We will improve the clarity of writing, our motivation, and the description of our method. We will also add 2 new experiments.**
>
> Thanks for your review. Following reviewers suggestions, we will make the following major changes in our next draft:
> - We will clearly explain that our main motivation is to improve over the method of Johnson et.al., 2016.
> - We will clarify the description, especially the conjugacy requirements, and add detailed discussion on the applicability and limitations of our approach.
> - We will clean-up the description of our algorithm in Section 4.
> - We will add an experiment on non-conjugate mixture model to demonstrate the generality of our approach. We will add a larger experiment for clustering MNIST digits.
> ----------
> DETAILED COMMENTS
> Reviewer: “One of the main aspects it seems is generality, but the encodings are specific to 2 types PGMs. Can this be generalized to arbitrary PGM structures? How about cases when computing Z is intractable?”
>
> Response: We agree that the write-up is not clear. We will improve this. Our method can handle arbitrary PGM structure in the model similar to the method of Khan and Lin 2017. The inference network however is restricted to cases where Z is tractable.  Our method therefore simplifies inference by choosing an inference network which has a simpler form than the original model.
> ---------
> Reviewer: “Regarding the experiments, as mentioned in the paper the evaluation is performed on two fairly small scale datasets.”
>
> Response: We agree with your point. Our comparisons are restricted because the existing implementation of SVAE baseline did not scale to large problems. We will add two more experiments as promised above.

---

> ### Author Response · Authors · 2018-01-05
> **Changes in the Revised Version**
>
> We have made the following changes in the revised version:
> - Introduction is modified to show that our method is an improvement over Johnson et. al.’s method, and it builds upon Khan and Lin’s method.
> - Section 2 modified to clearly show the issues with Johnson et.al.’s method.
> - Section 3 modified to clarify the conjugacy requirements of inference network. We have added many illustrative examples.
> - Section 4 modified to simplify the algorithm description. We have added a pseudo-code.
> - In Section 5 we added a new result on Student’s-t mixture model.

---

### Decision · Program_Chairs · 2018-01-29
**ICLR 2018 Conference Acceptance Decision**

**Decision:**

Accept (Poster)

**Comment:**

Thank you for submitting you paper to ICLR. The paper presents a general approach for handling inference in probabilistic graphical models that employ deep neural networks. The framework extends Jonhson et al. (2016) and Khan & Lin (2017). The reviewers are all in agreement that the paper is suitable for publication. The paper is well written and the use of examples to illustrate the applicability of the methods brings great clarity. The experiments are not the strongest suit of the paper and, although the revision has improved this aspect, I would encourage a more comprehensive evaluation of the proposed methods. Nevertheless, this is a strong paper.